# External Validation of a Population-Based Prediction Model for High Healthcare Resource Use in Adults

**DOI:** 10.3390/healthcare8040537

**Published:** 2020-12-04

**Authors:** Laura C. Rosella, Kathy Kornas, Joykrishna Sarkar, Randy Fransoo

**Affiliations:** 1Dalla Lana School of Public Health, University of Toronto, Toronto, ON M5T 3M7, Canada; kathy.kornas@utoronto.ca; 2ICES, Toronto, ON M4N 3M5, Canada; 3Manitoba Centre for Health Policy, University of Manitoba, Winnipeg, MB R3E 3P5, Canada; joykrishna_sarkar@cpe.umanitoba.ca (J.S.); randy_fransoo@cpe.umanitoba.ca (R.F.)

**Keywords:** healthcare utilization, high resource users, prediction

## Abstract

Predicting high healthcare resource users is important for informing prevention strategies and healthcare decision-making. We aimed to cross-provincially validate the High Resource User Population Risk Tool (HRUPoRT), a predictive model that uses population survey data to estimate 5 year risk of becoming a high healthcare resource user. The model, originally derived and validated in Ontario, Canada, was applied to an external validation cohort. HRUPoRT model predictors included chronic conditions, socio-demographics, and health behavioural risk factors. The cohort consisted of 10,504 adults (≥18 years old) from the Canadian Community Health Survey in Manitoba, Canada (cycles 2007/08 and 2009/10). A person-centred costing algorithm was applied to linked health administrative databases to determine respondents’ healthcare utilization over 5 years. Model fit was assessed using the c-statistic for discrimination and calibration plots. In the external validation cohort, HRUPoRT demonstrated strong discrimination (c statistic = 0.83) and was well calibrated across the range of risk. HRUPoRT performed well in an external validation cohort, demonstrating transportability of the model in other jurisdictions. HRUPoRT’s use of population survey data enables a health equity focus to assist with decision-making on prevention of high healthcare resource use.

## 1. Introduction

Canada has among the highest healthcare spending in the world, with total costs for 2018 estimated at CAD 253.5 billion or CAD 6839 per Canadian [1]. Across jurisdictions, healthcare spending is concentrated among a small proportion of the population [2]. From 2009 to 2011, 5% of healthcare users in Ontario, Canada accounted for 65% of healthcare costs [3]. Predictive tools that estimate new high cost users can inform cost reduction strategies. Existing healthcare utilization models that can be applied to new health settings, outside from where the model was originally developed, should demonstrate external validity [4,5]. Geographic validation (a type of external validation that examines how the model performs in individuals from a different health setting from where the model was originally developed) is especially suited for establishing the generalisability of a predictive model to health systems in other jurisdictions [6]. 

The High Resource User Population Risk Tool (HRUPoRT) is a population-based predictive 5 year model of adults who will become the top 5% of healthcare users based on self-reported health, sociodemographic, and health behavioural information that is routinely collected in population surveys [7]. HRUPoRT was developed in a cohort of 58,617 people from Ontario, Canada who responded to the 2007/08 Canadian Community Health Survey (CCHS) and was temporally validated in a cohort of 29,721 Ontarians in the 2009/10 CCHS [7]. The top 5% of healthcare users were calculated based on ranking individuals according to gradients of cost within the CCHS cohort using a person-centered costing methodology [7]. HRUPoRT is unique in that it was designed for use at the population and health system level, and the algorithm can be readily applied in other jurisdictions given that the model inputs are widely accessible in population survey data. The objective of this study was to establish HRUPoRT’s generalizability to other jurisdictions by geographically validating the Ontario derived HRUPoRT algorithm in another Canadian province, using self-reported risk factors from the Canadian Community Health Survey and high resource user status ascertained by administrative data from Manitoba’s health system. This type of geographic (cross-provincial) validation is important for demonstrating whether the HRUPoRT model can accurately predict high resource users in other jurisdictions, ensuring wide-scale generalizability of the tool.

## 2. Materials and Methods

### 2.1. Context and Setting

A prospective cohort study conducted in the province of Manitoba, Canada was used to externally validate the High Resource User Population Risk (HRUPoRT) model [7]. Manitoba has a single payer health insurance system that provides universal health coverage to its population of about 1.28 million residents, as of 2016 [8]. The study utilized linked population health surveys and health administrative data held and accessed at the Manitoba Centre for Health Policy. The study was approved by the research ethics boards at the University of Manitoba (#HS20593/H2017:093) and University of Toronto (#31967).

### 2.2. Data Sources

The study cohort was created by linking respondents from the combined 2007/2008 and 2009/2010 Canadian Community Health Surveys (CCHS) to provincial health administrative data. The CCHS is a cross-sectional survey administered by Statistics Canada that collects self-reported health related data and uses a probability sample and weighting system that is representative of 98% of the Canadian population aged 12 years and older living in private dwellings. Excluded from the CCHS sampling frame are individuals living in First Nation communities, full-time members of the Canadian Forces, individuals living in long-term care institutions, and residents of certain remote regions. The detailed survey methodology of the CCHS is described elsewhere [9]. Cycles of the CCHS were combined using the pooled approach [10].

Data on health services utilized for the 5 years after the CCHS interview date were obtained from the health administrative databases that are linked through a unique anonymized personal health identification number (PHIN). The administrative databases included Medical Services Data, Discharge Abstract Database, Case Mix Grouper data, National Rehabilitation Reporting System, Long Term Care Utilization database, and Drug Program Information Network. A description of these databases is available in Appendix A.

### 2.3. Participants

The external validation cohort was created using identical eligibility criteria as were applied in the original development study. The cohort consisted of CCHS respondents aged 18 years and older, who had a valid health card and agreed to have their survey responses linked to the provincial health administrative data. For individuals that appeared in multiple CCHS cycles, only data collected from their first CCHS interview were used. After exclusions, the cohort consisted of 10,504 respondents from the pooled 2007–2010 CCHS.

### 2.4. High Resource User Outcome

Healthcare utilization costs for each of the 5 years following the CCHS interview date were computed by applying a person-centered costing approach to the linked health administrative databases [11]. The costing algorithm estimated individual healthcare costs for utilization of physician services, inpatient hospitalizations, rehabilitation, long-term care, same day surgery, and pharmaceuticals. Individuals were ranked in each year according to the total annual per-person healthcare expenditures, with high resource users defined as the top 5% of users in any given year.

Due to data availability, costs did not include health services for complex continuing care, home care, emergency department visits, inpatient mental health, and assisted medical devices, as were included in the ascertainment of high resource user status in the original HRUPoRT development cohort [7]. As well, due to differences in provincial drug plans (income-based drug coverage in Manitoba versus age-based drug coverage in Ontario for people aged 65 and older), a wider population coverage of drug costs was included in the Manitoban cohort compared to the original development cohort.

### 2.5. Predictors

The HRUPoRT model consists of 12 predictor variables; these were obtained from the CCHS, as of the respondent’s interview date. The process used to select the variables for the HRUPoRT model has been described elsewhere [7]. Demographic and socioeconomic predictors included sex, age group (<30, 30–39, 40–49, 50–59, 60–69, 70–79, 80+), household income quintile, ethnicity (white, non-white), immigrant status (immigrant <10-years, immigrant ≥10 years, non-immigrant), and food security (food secure, food insecure). Health status predictors included history of a chronic condition and self-reported general health (excellent/very good/good, fair, poor). Body mass index (BMI) was categorized as <18.5 kg/m^2^, 18.5–24.9 kg/m^2^, 25.0–29.9 kg/m^2^, 30.0–34.9 kg/m^2^, 35.0–39.9 kg/m^2^, and ≥40.0 kg/m^2^. Health behavioural predictors included smoking status (current light smoker, current heavy smoker, former light smoker, former heavy smoker, non-smoker), physical activity quartile, and alcohol consumption in the past 12 months (heavy drinker, moderate drinker, light drinker and non-drinker). 

All predictors and categorizations were defined using the same CCHS questions as applied in the original development cohort [7]. An exception to this was alcohol consumption, in which due to provincial differences in CCHS questions related to alcohol use, categories were defined based on how often alcohol was consumed in the past 12 months as opposed to the number of drinks consumed per week in the past 12 months. Specifically, alcohol consumption was defined as heavy drinker (drinks 1–6 times a week or everyday and binges once or more than once a week), moderate drinker (drinks once a week and binges 1–3 times a month; or, drinks 2–3 times a week and binges less than 3 times a month; or, drinks 4–6 times a week or everyday and binges less than 3 times a month or never), light drinker (drinks 1–3 times a week and never binges), non-drinker (drinks less than weekly or no alcohol consumption in the last 12 months). The revised definition for alcohol categories was informed by alcohol consumption definitions used in previous literature [12]. 

### 2.6. External Validation of the Model

All predictor variables were centered to the mean to account for differences between populations and the distribution of risk factors. Missing values for predictors were maintained as separate categories. For each individual in the cohort, the predicted 5 year probability of high resource user status was computed by using model coefficients that were derived from the original development cohort, but which were updated from the previously published model using mean-centered risk factor variables (see Appendix A
Table A1). The probabilities were computed using the formula: probability = exp(logit)/(1 + (exp(logit))), in which the logit is the sum of the regression coefficients multiplied by their respective predictor variable values.

The predicted probabilities were used to evaluate the model’s performance relative to discrimination and calibration. Discrimination refers to the ability of the model to differentiate between those who will, and those will not become a high resource user. Discrimination was measured using the c-statistic, which is identical to the area under the receiver operating characteristic curve [13]. Calibration was assessed by grouping the observations into deciles of risk and comparing the agreement between predicted risk and observed high resource user outcomes. Calibration was visually assessed by observing the calibration plot across deciles of high resource user risk [13]. The overall performance of the models was assessed using the likelihood ratio (R^2^), which measures the variation explained by a model, and the Brier score, which measures the accuracy of predictions by calculating the squared difference between outcome and predictions [13]. 

### 2.7. General Statistical Analyses

All estimates were weighted using sampling survey weights provided by Statistics Canada to account for the complex survey design of the CCHS and to be representative of the provincial population. Confidence intervals were estimated using bootstrap weights applied using the balanced repeated replication approach for standard error estimation. All statistical analyses were conducted using SAS, version 9.4 (SAS institute, Cary, NC, USA).

## 3. Results

Baseline characteristics of the external validation cohort are summarized in Table 1. At the end of the 5 year follow-up period, 10.9% (*n* = 1145) of respondents became a high resource user. In the original development cohort, 6.0% of respondents became a high resource user [7]. The difference in magnitude may be due to differences in the provincial distribution of immigrant status and non-white ethnicity, which are protective of high resource user risk. The cohort had a similar age, sex, and income distribution compared to individuals in the original development cohort [7]. In addition, the presence of a chronic condition and the distribution of health and behavioural characteristics (BMI, smoking, physical activity, alcohol consumption) were similar in the current and original development cohort. Major cohort differences were that the Manitoba cohort had fewer immigrants (11.0% vs. 23.7%) and individuals of non-white ethnicity (9.9% vs. 21.7%), compared to the original Ontario development cohort. 

In the external validation cohort, the HRUPoRT model demonstrated strong discriminative-ability (c-statistic = 0.8264). This was similar to discrimination observed in the original development (c-statistic = 0.8213) and validation (c-statistic = 0.8171) cohorts [7]. The R^2^ statistic was 0.1408 and the Brier score was 0.0835, indicating appropriate overall model performance.

Figure 1 shows the calibration plot for observed and predicted high resource user cases across decile risk groups. The model was well calibrated across the spectrum of risk, with the exception of an underprediction of 28% in decile two, 40% in decile four, and an overprediction of 33% in decile one. It is possible that other risk factors not captured by the model are more predictive of high resource user risk for people in these lower decile risk groups. Otherwise, differences between observed and predicted cases were in the range of 7% to 14%. 

## 4. Discussion

This study externally validated a previously developed population-based model for predicting the five-year risk of high healthcare resource utilization [7]. We demonstrated that the HRUPoRT had good discrimination and was well calibrated throughout the range of risk in an external population that was distinguished by geography from the original derivation and validation cohorts. The HRUPoRT can be applied to identify priority populations and inform population-level prevention of high resource users and reduction in cost to the healthcare system in other jurisdictions. HRUPoRT’s ability to predict high users could also facilitate a process by which health ministries could identify clusters of residents at high risk in the community and target interventions to prevent high healthcare use.

HRUPoRT is uniquely designed for use on routinely collected population surveys, which contain social determinants of health information, enabling a focus on health equity in decision-making. Socioeconomic factors, including income, food security, and life satisfaction, are known to be associated with high resource use [14,15,16]. These data are not readily collected in electronic medical records or health administrative data, which are common data sources for other existing healthcare utilization models [17,18]. Population health surveys are widely available across countries and regions, and importantly, tools that can make use of this data can help identify clusters of health disparities in communities.

A limitation of this study was the difference in the measurement of HRU status, as compared with the development study. Specifically, due to data availability, fewer healthcare services were included in calculating healthcare costs for individuals in the validation cohort, including the absence of costs for the emergency department and home care. Although this study used fewer types of health services to calculate healthcare costs, the characteristics of HRUs identified in our cohort were similar to those reported in the original development study. Furthermore, due to the sampling frame of the CCHS, the model’s performance among populations living in institutional facilities and Indigenous people living in First Nation communities is unknown. Finally, the HRUPoRT does not include clinical variables related to illness level, which is a key determinant of health service utilization [19]. This was due to our aim to build a model that could run solely on routinely collected population health survey data to enable wide-scale use. It is possible that the integration of clinical variables, and other risk factors that may be more predictive of HRU risk for people in the lower risk deciles, could improve the predictive performance of the model. Future research focused on improving predictive performance could explore updating the model using linkages of population health surveys and administrative data, noting that including more variables that are not easily accessible to planners could reduce usability. Nonetheless, it is noteworthy that the HRUPoRT accurately predicts risk in high-risk decile groups, which is a key purpose of the model. 

Containing healthcare spending has been identified by governments in multiple health systems as a top priority. The HRUPoRT considers the upstream determinants of high-cost users, which is useful for informing the design of different HRU prevention strategies at the community level. The HRUPoRT is intended to be used by health system planners and decision-makers as an aid in population-based planning. The tool’s relevance has been demonstrated by its integration into routine practice at a major public health unit in Ontario to understand the determinants of high resource use, which is a core focus of public health [20]. The HRUPoRT can inform how risk is distributed in communities and help identify which population groups would benefit from targeted interventions. Population surveys that are suitable for HRUPoRT application in other jurisdictions include The National Health Interview Survey (NHIS) in the United States, National Health Survey in England, New Zealand Health Survey, among others. Demonstrating performance of prediction models in various settings is an important step to ensure validity when implemented in other regions. Future research can focus on the application of the HRUPoRT for modelling HRU prevention strategies in different jurisdictions.

## 5. Conclusions

This study has demonstrated the geographical validation and resulting predictive performance of the HRUPoRT, offering evidence to support the transportability of the model in other jurisdictions. The HRUPoRT’s use of population health survey data enables a focus on health equity to assist with decision-making on high healthcare resource use prevention. 

## Figures and Tables

**Figure 1 healthcare-08-00537-f001:**
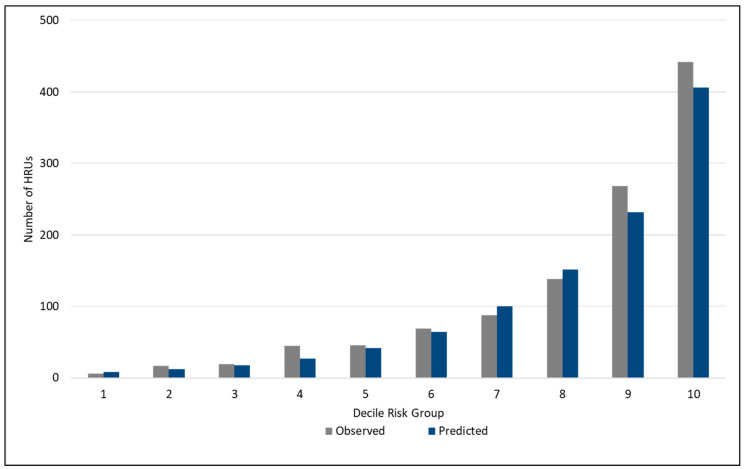
Calibration plot for the High Resource User Population Risk Tool (HRUPoRT) model in the external validation cohort.

**Table 1 healthcare-08-00537-t001:** Weighted baseline characteristics of the external validation cohort ^1^.

Risk Factor	HRU ^2^, Top 5% % (95% CI) *n* = 1145	Non-HRU, Bottom 95% % (95% CI) *n* = 9359
**Sex**		
Male	49.7 (44.5, 54.8)	49.3 (48.5, 50.1)
Female	50.3 (45.2, 55.5)	50.7 (49.9, 51.5)
**Age group**		
<30	3.5 (1.8, 5.2)	22.9 (22.1, 23.6)
30–39	5.4 (2.8, 8.0)	18.0 (17.1, 19.0)
40–49	10.7 (7.3, 14.1)	19.5 (18.0, 21.0)
50–59	17.2 (12.2, 22.1)	17.8 (16.5,19.0)
60–69	21.1 (16.3, 25.8)	12.3 (11.3, 13.4)
70–79	18.2 (15.4, 21.0)	6.6 (6.1, 7.1)
≥80	23.9 (20.5, 27.3)	2.8 (2.5, 3.2)
**Ethnicity**		
White	86.3 (82.0, 90.6)	79.3 (77.6, 80.9)
Non-white	Suppressed	9.9 (8.5, 11.3)
Missing	8.0 (5.5, 10.5)	10.8 (9.8, 11.9)
**Immigrant status**		
Non-immigrant	84.0 (79.6, 88.4)	84.4 (82.9, 85.8)
Immigrant (<10 years)	Suppressed	4.5 (3.7, 5.3)
Immigrant (≥10 years)	15.4 (11.0, 19.9)	11.0 (9.8, 12.2)
Missing	Suppressed	0.2 (0.1, 0.3)
**Household income**		
Q1 (lowest)	24.7 (20.5, 28.8)	17.4 (15.9, 18.9)
Q2	22.5 (18.7, 26.3)	17.2 (15.8, 18.6)
Q3	16.8 (12.5, 21.2)	18.0 (16.4, 19.6)
Q4	11.9 (7.8, 16.0)	18.0 (16.6, 19.5)
Q5 (highest)	8.7 (6.1, 11.4)	18.8 (17.4, 20.1)
Missing	15.4 (11.3, 19.5)	10.6 (9.5, 11.6)
**Food Security**		
Food Secure	92.4 (89.3, 95.5)	92.7 (91.8, 93.7)
Food Insecure	5.0 (2.6, 7.4)	5.9 (5.0, 6.7)
Missing	Suppressed	1.4 (1.0, 1.8)
**Chronic Condition**		
Yes	84.8 (81.0, 88.6)	56.5 (54.8, 58.2)
No	13.6 (10.2, 17.1)	37.5 (35.7, 39.2)
Missing	Suppressed	6.1 (5.2, 6.9)
**General Health**		
Excellent/very good	30.2 (25.4, 35.0)	59.5 (57.7, 61.2)
Good	33.4 (28.2, 38.4)	29.8 (28.2, 31.5)
Fair	24.7 (20.9, 28.5)	8.8 (7.8, 9.7)
Poor	11.7 (8.5, 14.9)	1.9 (1.5, 2.3)
Missing	Suppressed	Suppressed
**Body Mass Index**		
<18.5 kg/m^2^	Suppressed	2.0 (1.4, 2.5)
18.5–24.9 kg/m^2^	32.0 (27.4, 36.5)	38.4 (36.8, 40.0)
25.0–29.9 kg/m^2^	34.6 (29.6, 39.7)	34.4 (32.8, 36.0)
30.0–34.9 kg/m^2^	15.8 (11.8, 19.8)	13.5 (12.3, 14.7)
35.0–39.9 kg/m^2^	4.2 (2.3, 6.2)	4.6 (3.9, 5.3)
≥40.0 kg/m^2^	Suppressed	2.0 (1.5, 2.5)
Missing	7.6 (5.0, 10.1)	5.1 (4.3, 5.9)
**Smoking Status**		
Heavy smoker	7.45 (3.4, 11.5)	3.1 (2.5, 3.7)
Light smoker	15.4 (12.0, 18.8)	18.9 (17.6, 20.2)
Former heavy smoker	13.1 (9.0, 17.1)	7.0 (6.2, 7.9)
Former light smoker	22.8 (18.6, 27.0)	16.2 (14.9, 17.5)
Non-smoker	37.0 (32.3, 41.7)	51.2 (49.4, 53.0)
Missing	4.3 (2.8, 5.8)	3.5 (2.9, 4.2)
**Physical activity**		
Q1 (lowest)	35.4 (30.5, 40.3)	24.1 (22.5, 25.6)
Q2	26.1 (21.7, 30.4)	23.2 (21.9, 24.6)
Q3	19.9 (15.5, 24.3)	25.1 (23.5, 26.6)
Q4 (highest)	13.6 (10.3, 17.0)	25.9 (24.3, 27.4)
Missing	5.0 (2.6, 7.3)	1.7 (1.2, 2.3)
**Alcohol consumption**		
Heavy drinker	4.9 (2.7, 7.0)	8.0 (6.9, 9.1)
Moderate drinker	10.7 (8.1, 13.3)	19.3 (17.9, 20.7)
Light drinker	12.7 (8.8, 16.6)	14.7 (13.6, 15.8)
Non-drinker	71.3 (66.6, 76.1)	57.4 (55.7, 59.1)
Missing	Suppressed	0.6 (0.4, 0.8)

^1^ Numbers are weighted percentages using bootstrap weights by Statistics Canada. Values are suppressed due to small cell sizes (<6 individuals). ^2^ Abbreviations: HRU, high resource user.

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
