# Peer review of "External Validation of a Population-Based Prediction Model for High Healthcare Resource Use in Adults"

_healthcare, 2020, doi:10.3390/healthcare8040537_

Round 1

Reviewer 1 Report

First, I would like to congratulate the authors for carrying out their research focusing on this interesting topic. Nevertheless, there are required some changes:

  1. The affiliation of the authors is missing the data such as the post code. 
  2. The introduction could include more information of the health are spending in Canada and how or when the High Resource User Population Risk Tool was created, despite having the reference five. Also, the first time that the reference [5] appears should be before the dot. 
  3. The number or code of Research Ethics Boards at the University of Manitoba and University of Toronto should be included in the manuscript. 
  4. The results are highly impressive but, I consider that the authors could highlight some interesting information since the table is massive. 
  5. Finally, the discussion and conclusions are well-written but its oddly include the relevance or practical implications. 

Reviewer 2 Report

This paper appears to be correlated with an earlier study listed below.  It is not immediately clear if the current study is sufficiently different enough to support a second paper on this topic. More explanation is needed on geographic validation.

Predicting High Health Care Resource Utilization in a Single-payer Public Health Care System: Development and Validation of the High Resource User Population Risk Tool

Please expand the discussion to present suggestions for future research.

There is more self-citing than usual, but this is a natural outcome given they paper is an application of their earlier research.

Reviewer 3 Report

it is well known that a relatively small proportion of the population account for a large proportion of healthcare costs. The study confirms this.

This article is well-done in terms of methods and conceptualization relative to the goal of generalization to other jurisdictions. As well as the analysis itself.

Use of the HRUPoRT to identify the top 5% pf users, linking to CCHS, as well as administrative data bases is a most important feature. 
The fact that not all healthcare services were not captured is not a major omission in my view.’

The 12 predictor variables are appropriate for the external validation of the model.  I would like to know if other predictor variables were considered and not used.   Were the predictors dictated by some underlying theoretical model or were these variables the only ones available for use?   I suggest that the authors examine a model such as the Behavioral Model of Health Services Utilization (Andersen & Newman and/or subsequent variations and use it to indicate where other possible variables should be included in future analysis in order to to achieve a larger R-square as well as a possible way to target interventions.

I find it interesting that 10.9% became high resource users.  Would the authors expect this to be generally true in other provinces?

Figure one is interesting is what it shows and what it does not show.  What are some possible reasons for the decile 1, 2 and 4 deviations?

While the paper is largely methodological, I would like to know, in light of Figure 1, what is needed to make the results have greater predictive  ability as well as suggesting some specific strategies for use of the model in the delivery of care to high cost users?
